# Recurrent intragenic rearrangements of *EGFR* and *BRAF* in soft tissue tumors of infants

Jenny Wegert[1], Christian Vokuhl[2], Grace Collord [3,4], Martin Del Castillo Velasco-Herrera [3], Sarah J. Farndon[3,5], Charlotte Guzzo [3], Mette Jorgensen[6], John Anderson[5,6], Olga Slater[6], Catriona Duncan[6], Sabrina Bausenwein[1], Heike Streitenberger[1], Barbara Ziegler[1], Rhoikos Furtwängler[7], Norbert Graf [7], Michael R. Stratton[3], Peter J. Campbell[3], David TW Jones[8,9], Christian Koelsche [10,11,12], Stefan M. Pfister[8,9,13], William Mifsud[6], Neil Sebire[5,6], Monika Sparber-Sauer[14], Ewa Koscielniak[14,15], Andreas Rosenwald[16,17], Manfred Gessler [1,17] & Sam Behjati[3,4]

Soft tissue tumors of infancy encompass an overlapping spectrum of diseases that pose unique diagnostic and clinical challenges. We studied genomes and transcriptomes of cryptogenic congenital mesoblastic nephroma (CMN), and extended our findings to five anatomically or histologically related soft tissue tumors: infantile fibrosarcoma (IFS), nephroblastomatosis, Wilms tumor, malignant rhabdoid tumor, and clear cell sarcoma of the kidney. A key finding is recurrent mutation of *EGFR* in CMN by internal tandem duplication of the kinase domain, thus delineating CMN from other childhood renal tumors. Furthermore, we identify *BRAF* intragenic rearrangements in CMN and IFS. Collectively these findings reveal novel diagnostic markers and therapeutic strategies and highlight a prominent role of isolated intragenic rearrangements as drivers of infant tumors.

[1] Theodor-Boveri-Institute/Biocenter, Developmental Biochemistry, University of Wuerzburg, 97074 Wuerzburg, Germany. [2] Kiel Pediatric Tumor Registry, Section of Pediatric Pathology, Department of Pathology, Christian Albrechts University, 24105 Kiel, Germany. [3] Wellcome Trust Sanger Institute, Hinxton, CB10 1SA, UK. [4] Department of Paediatrics, University of Cambridge, Cambridge, CB2 0QQ, UK. [5] UCL Great Ormond Street Institute of Child Health, London, WC1N 1EH, UK. [6] Great Ormond Street Hospital for Children NHS Foundation Trust, London, WC1N 3JH, UK. [7] Department of Pediatric Oncology and Hematology, Saarland University Hospital, 66421 Homburg, Germany. [8] Hopp Children's Cancer Center at the NCT Heidelberg (KiTZ), 69120 Heidelberg, Germany. [9] Department of Pediatric Neurooncology, German Cancer Research Center (DKFZ) and German Cancer Consortium (DKTK), 69120 Heidelberg, Germany. [10] Clinical Cooperation Unit Neuropathology, German Cancer Research Center (DKFZ), 69120 Heidelberg, Germany. [11] Department of Neuropathology, Institute of Pathology, Heidelberg University Hospital, 69120 Heidelberg, Germany. [12] Department of General Pathology, Institute of Pathology, Heidelberg University Hospital, 69120 Heidelberg, Germany. [13] Department of Pediatric Hematology and Oncology, Heidelberg University Hospital, 69120 Heidelberg, Germany. [14] Klinikum Stuttgart—Olgahospital, Stuttgart Cancer Center, Zentrum für Kinder-, Jugend- und Frauenmedizin, Pediatrics 5 (Oncology, Hematology, Immunology), 70174 Stuttgart, Germany. [15] Department of Pediatric Hematology and Oncology, Children's Hospital, 72076 Tübingen, Germany. [16] Institute of Pathology, University of Wuerzburg, 97080 Wuerzburg, Germany. [17] Comprehensive Cancer Center Mainfranken, University of Wuerzburg, 97078 Wuerzburg, Germany. These authors contributed equally: Jenny Wegert, Christian Vokuhl, Grace Collord, Martin Del Castillo Velasco-Herrera. These authors jointly supervised this work: Manfred Gessler, Sam Behjati. Correspondence and requests for materials should be addressed to M.G. (email: gessler@biozentrum.uni-wuerzburg.de) or to S.B. (email: sb31@sanger.ac.uk)

Many childhood tumors show a predilection for specific developmental stages. Tumors that predominantly occur in infancy include congenital mesoblastic nephroma (CMN), which accounts for 4% of all childhood renal malignancies and the majority of those diagnosed in children under 6 months of age[1,2]. CMN is classified histologically into classical, cellular, and mixed subtypes based primarily on degree of cellularity and mitotic activity[3]. The cellular variant is characterized by a sarcoma-like diffuse hypercellular morphology, whereas classical CMN is composed of less proliferative spindle cells[3]. Cellular CMN is driven by rearrangements involving the tropomyosin receptor kinase (TRK) gene NTRK3, most commonly a t(12;15)(p13;q25) reciprocal translocation with the ETV6 transcription factor[4,5]. Less frequent somatic aberrations include trisomies of chromosomes 8, 11, 17, and 20[6,7] and rarer TRK fusions, involving NTRK1, NTRK2, or NTRK3[8]. By contrast, the genetic changes underpinning the classical variant, accounting for >30% of cases, are unknown[9]. Cellular CMN shares its genetic and morphological hallmarks with infantile fibrosarcoma (IFS), a spindle cell tumor typically arising in the soft tissues of the extremities or abdomen[5,9,10].

Standard treatment for CMN and IFS is complete surgical resection[9–11]. In the case of IFS, local control frequently requires cytotoxic chemotherapy[10,11]. The role for up-front chemotherapy in CMN is less clear[9]. Recently, a phase I/II clinical trial of a selective TRK inhibitor, larotrectinib, reported high response rates in diverse tumor types harboring TRK gene fusions, including IFS and other soft tissue tumors of infancy[12]. Morbidity and infrequent death result from tumor recurrence or from treatment-related complications[9–11].

Here, we investigated the genetic basis of CMN and IFS lacking the canonical NTRK3-ETV6 fusion gene. We identify oncogenic rearrangements in MAPK signaling genes across all cases interrogated by unbiased sequencing, notably therapeutically tractable intragenic rearrangements in EGFR and BRAF.

## Results

**Overview of the genomic landscape of CMN.** To identify the genetic basis of cryptogenic CMN, we first applied whole genome and transcriptome sequencing to a discovery cohort of ten classical CMN lacking an NTRK3 fusion (Supplementary Data 1). Somatic variants were identified by comparing tumor and matched peripheral blood sequences (see Methods). The genomic landscape was universally quiet, with a low burden of point mutations (median of 45 substitutions and 9 insertions or deletions per genome; Supplementary Data 2). The predominant mutational signatures, as defined by the trinucleotide context of substitutions, were the ubiquitous signatures 1 and 5[13]

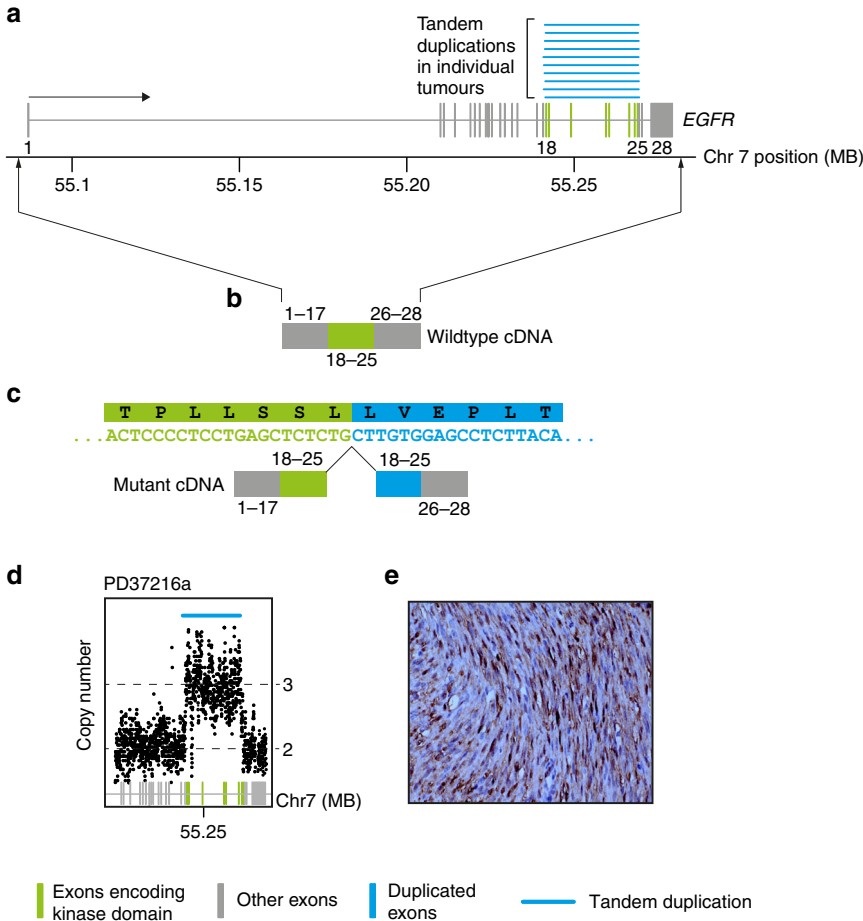

**Fig. 1** EGFR internal tandem duplication. **a** The genomic footprint of EGFR is depicted with exons represented by gray and green vertical lines. Green exons encode the kinase domain. Blue lines superiorly show the tandem duplications found in the discovery cohort of ten congenital mesoblastic nephroma of classical histology. **b** Schematic of the wild-type transcript. **c** Schematic of the fusion transcript annotated with cDNA sequence of rearrangements (sense orientation) and protein translation. **d** Intragenic copy number of EGFR showing focal amplification over the kinase domain (x-axis: genomic coordinate; y-axis: copy number derived from coverage). **e** Representative phospo-ERK immunohistochemistry

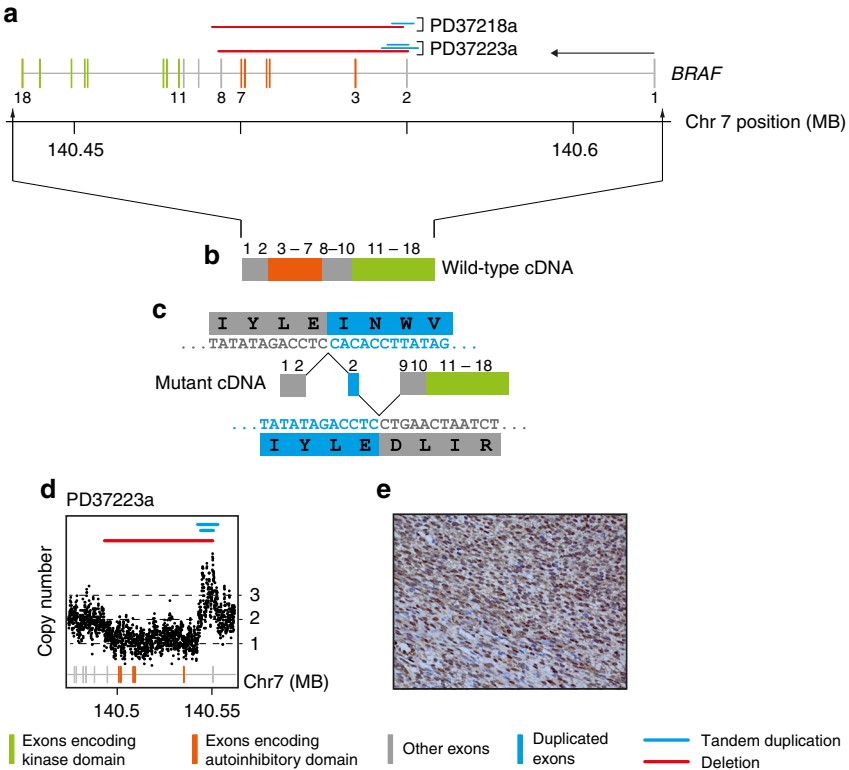

**Fig. 2** Internal *BRAF* deletion. **a** The genomic footprint of *BRAF* is depicted with exons represented by gray, green, and orange vertical lines. Green and orange exons encode the kinase domain and conserved region 1, respectively. Horizontal lines above exons demarcate rearrangements (blue: tandem duplication; red: deletion). **b** Outline of wild-type transcript. **c** Outline of fusion transcript with cDNA sequence of rearrangements (sense orientation) with translation. **d** Intragenic copy number of *BRAF* (x-axis: genomic coordinate; y-axis: copy number derived from coverage). **e** Representative phospho-ERK immunohistochemistry

(Supplementary Fig. 1). Copy number changes and structural rearrangements were likewise scarce (Supplementary Fig. 2).

**Internal tandem duplication of the *EGFR* kinase domain in CMN.** Annotating all cases for potential oncogenic variants revealed a single intragenic, in-frame internal tandem duplication (ITD) of the *EGFR* kinase domain in all ten tumors (Table 1; Fig. 1; Supplementary Data 3). The breakpoints clustered in a narrow genomic window around the kinase domain of *EGFR* encoded in exons 18−25 (Fig. 1a). This rearrangement is rarely observed in several other tumor types including in glioma and in lung adenocarcinoma, and confers sensitivity to a targeted EGFR inhibitor, afatinib[14]. We validated all rearrangements by genomic copy number analysis and reconstruction of cDNA reads spanning the breakpoint junction (Fig. 1; see Methods). Of note, the same mutant cDNA junction sequence was found in every case, irrespective of the genomic location of breakpoints. A search for additional known or novel driver variants revealed no further plausible candidates in any of the *EGFR*-mutant tumors. We next extended this investigation to seven non-classical CMN lacking an *NTRK3* fusion, including four mixed cellularity cases and three cellular tumors (Table 1; Supplementary Data 1). Two of the four mixed cellularity tumors surveyed also harbored an *EGFR*-ITD. Of note, for one child with *EGFR*-ITD-positive mixed cellularity CMN (PD37214), both primary tumor and recurrence were studied, with no additional driver events apparent at relapse.

**BRAF rearrangements in CMN and IFS.** A further striking finding was the discovery of mutations in the *BRAF* oncogene in 2/3 cellular histology CMNs. *BRAF* fusions have been implicated in a minority of IFS but not in CMN[15]. In both cases the *BRAF*

rearrangement involved a compound deletion of conserved region 1 (CR1) and tandem duplication of exon 2 (Fig. 2; Table 1; Supplementary Data 3). CR1 encompasses the negative regulatory Ras-binding domain (RBD), loss of which is predicted to generate a constitutively active form of BRAF[16,17]. Mutated tumors displayed intense staining of phosphorylated ERK by immunohistochemistry, consistent with activated signaling downstream of BRAF (Figs. 1e and 2e). A further tumor harbored the *KIAA1549-BRAF* fusion, a molecular hallmark of a childhood brain tumor, pilocytic astrocytoma[18,19]. This fusion likewise results in loss of the N-terminal portion of the BRAF protein containing the RBD[17,18].

**Other TRK fusions in CMN.** The remaining two cases of CMN interrogated by whole genome and transcriptome sequencing were accounted for by gene fusions involving *NTRK1*, an alternate kinase of the TRK family of protein kinases: *TPR-NTRK1* and *LMNA-NTRK1*. Both of these fusions have been observed in IFS and rarely in adult cancers, but not, to our knowledge, in CMN[20–23] (Table 1). Hence, every cryptogenic CMN interrogated by whole-genome sequencing contained an oncogenic rearrangement in *BRAF*, *EGFR*, or *NTRK1*, all of which encode kinases involved in MAPK signaling and are amenable to inhibition with existing drugs[9,12,14,17,24].

**EGFR-ITD distinguishes CMN from other childhood renal tumors.** To validate and extend our findings, we screened IFS and a range of childhood renal tumors for *EGFR*-ITD, *BRAF*-ID, and *ETV6-NTRK3* using PCR. Tumor types included additional cases of CMN (n = 63), IFS (n = 26), Wilms tumor (n = 208), clear cell sarcoma of the kidney without *BCOR* rearrangements (n = 20), malignant rhabdoid tumor (n = 3), and nephroblastomatosis

**Table 1 Rearrangements in infant soft tissue tumors**

| Assay | Tumor type | Subtype | Total | EGFR-ITD | BRAF-ID | BRAF-ID + ETV6-NTRK3 | ETV6-NTRK3 | KIAA1549-BRAF | LMNA-NTRK1 | EML4-NTRK3 | TPR-NTRK1 |
|---|---|---|---|---|---|---|---|---|---|---|---|
| WGS + mRNA sequencing | CMN | Cellular | 3 | 0 | 2 | 0 | 0 | 0 | 1 | 0 | 0 |
| | | Classical | 10 | 10 | 0 | 0 | 0 | 0 | 0 | 0 | 0 |
| | | Mixed | 4 | 2 | 0 | 0 | 0 | 1 | 0 | 0 | 1 |
| | IFS | — | 1 | 0 | 0 | 0 | 0 | 0 | 0 | 1 | 0 |
| PCR for EGFR-ITD, BRAF-ID and ETV6-NTRK3 | CMN | Cellular | 17 | 2 | 0 | 0 | 13 | – | – | – | – |
| | | Classical | 35 | 20 | 0 | 0 | 0 | – | – | – | – |
| | | Mixed | 11 | 9 | 0 | 0 | 0 | – | – | – | – |
| | IFS | – | 26 | 0 | 1 | 2 | 16 | – | – | – | – |
| | WT | – | 208 | 0 | 0 | 0 | 0 | – | – | – | – |
| | CCSK[a] | – | 20 | 0 | 0 | 0 | 0 | – | – | – | – |
| | MRT | – | 3 | 0 | 0 | 0 | 0 | – | – | – | – |
| | NB | – | 12 | 0 | 0 | 0 | 0 | – | – | – | – |

CMN congenital mesoblastic nephroma, IFS infantile fibrosarcoma, WT Wilms tumor, CCSK clear cell sarcoma of the kidney, MRT malignant rhabdoid tumor, NB nephroblastomatosis, WGS whole genome sequencing, mRNA messenger RNA, PCR polymerase chain reaction
[a]Negative for BCOR rearrangement

(n = 12; Table 1; Supplementary Data 1). EGFR-ITD was most prevalent in classical and mixed cellularity CMN, though was also found in cellular CMN (2/17 cases). The frequency of EGFR rearrangement in classical tumors was lower in the validation cohort (20/35 cases) than in the initial discovery cohort (10/10 cases). None of the IFS cases, nor other childhood kidney tumors, harbored EGFR-ITD. However, we encountered three cases of IFS with intragenic BRAF deletions. Remarkably, in two cases BRAF-ID co-occurred with NTRK3 fusions, the disease-defining mutation of IFS. We were unable to accurately estimate relative allele frequencies by nested PCR (see Methods). Hence, it is possible that both fusions co-exist within the same clone or represent independent clones that evolved in parallel within the same tumor.

## Discussion

In this exploration of infant tumors we identify ITD of the EGFR kinase domain that delineates a genetic subgroup of CMN transcending histological subtypes. Additionally, we report a novel rearrangement of BRAF present in both cellular CMN and IFS. These mutations represent diagnostic markers that can be readily integrated into routine clinical practice. Furthermore, EGFR and BRAF emerge as therapeutic targets, which may be exploited in certain clinical situations, e.g., large surgically intractable tumors, disease recurrence or metastases.

It is noteworthy that an oncogenic mutation was identified in every tumor that we studied by whole-genome sequencing. Of these, 78% harbored either EGFR-ITD or BRAF-ID, while the remaining 22% presented with non-canonical mutations involving BRAF, NTRK1, or NTRK3. This suggests that less recurrent rearrangement variants, albeit implicated in the same signaling circuity, may elude detection by targeted diagnostic assays. Moreover, our results indicate that a subset of tumors harbor multiple drivers with important implications for targeted therapy efforts. The finding of co-mutation of NTRK3 and BRAF in IFS raises the possibility of intrinsic resistance of some tumors to TRK inhibition, regardless of whether these mutations occur in the same clone or in independent competing clones. This finding is pertinent to clinical trials of TRK inhibitors in CMN and IFS[12]. In this vein a structurally similar BRAF fusion transcript, albeit without duplication of exon 2, has recently been implicated as a mechanism of resistance to certain BRAF/MEK inhibitors[16,17]. These considerations underscore the need for adequate genomic profiling in order to match patients to the most appropriate basket studies and to enable meaningful interpretation of treatment responses. Therefore, we would advocate extending the diagnostic work-up of refractory or relapsed CMN and IFS to whole genome sequencing, particularly in the context of clinical trials.

Biologically our findings draw further parallels between CMN and IFS. We identify BRAF and NTRK1 as additional cancer genes operative in both malignancies, substantiating the view that these diagnoses represent variants on the same disease spectrum converging on aberrant RAS-RAF-MEK-ERK signaling[5,8,9]. Furthermore, in the wider context of the childhood cancer genome, our findings add to the growing body of studies that identify short distance intragenic rearrangements as a dominant source of oncogenic mutations in otherwise quiet genomes. We note the parallel between CMN, clear cell sarcoma of the kidney and low-grade glioma that are in large part driven by ITDs often involving kinase domains, mostly as isolated driver events[18,25–29]. Furthermore, even in acute myeloid leukemia, where FLT3-ITD is a recurrent driver event in adult disease, childhood AML demonstrates a distinct structural variant profile enriched for focal chromosomal gains and losses[30]. We can only speculate on the biological significance of this parallel which may allude to specific mutational mechanisms operative during discrete stages of human development.

## Methods

**Patient samples**. All tissue samples were obtained after gaining written informed consent for tumor banking and future research from the patient (or their guardian) in accordance with the Declaration of Helsinki and appropriate national and local ethical review processes. German tissue samples were obtained from the following studies: SIOP93-01/GPOH and SIOP2001/GPOH (Ethikkommission der Ärztekammer des Saarlandes reference numbers 23.4.93/Ls and 136/01), the PTT2.0 study (Medical Faculty Heidelberg ethics reference number S-546/2016), the CWS trials CWS-96 and CWS-2002P (Universitätsklinikum Tübingen Medizinische Fakultät ethics approval, reference numbers 105/95 and 51/2003) and the SoTiSaR registry (ethics approval reference 158/2009B02). UK patients were enrolled under ethics approval from National Research Ethics Service Committee East of England, Cambridge Central (reference 16/EE/0394). Use of UK archival material was approved by the National Research Ethics Service Committee London Brent (reference 16/LO/0960). Additional tissue was obtained from the UK Children's Cancer and Leukaemia Group tissue bank.

**Sequencing**. Tumor DNA and RNA were extracted from fresh frozen tissue that had been reviewed by reference pathologists. Normal tissue DNA was derived from blood samples. Whole genome sequencing was performed by 150-bp paired-end sequencing on the Illumina HiSeq X platform. We followed the Illumina no-PCR library protocol to construct short insert libraries, prepare flowcells, and generate clusters. Coverage was at least 30×. Messenger RNA was enriched by polyA-

selection and sequenced on an Illumina HiSeq 2000 (paired end, 75-bp read length). DNA and RNA sequencing reads were aligned to the GRCh 37d5 reference genome using the Burrows−Wheeler transform (BWA-MEM)[31] and STAR (2.0.42)[32], respectively.

**Variant detection.** The Cancer Genome Project (Wellcome Trust Sanger Institute) variant calling pipeline was used to call somatic mutation and includes the following algorithms: CaVEMan (1.11.0)[33] for substitutions, an in-house version of Pindel (2.2.2; github.com/cancerit/cgpPindel)[34] for indels, BRASS (5.3.3; github.com/cancerit/BRASS) for rearrangements, and ASCAT NGS (4.0.0) for copy number aberrations[35]. RNA sequences were analyzed with an in-house pipeline (github.com/cancerit/cgpRna/wiki) which uses HTSeq[36] for gene feature counts, and a combination of TopHat-Fusion (v2.1.0)[37], STAR-fusion (v0.1.1)[32] and DeFuse (v0.7.0)[38] to detect expressed gene fusions. In addition to filters inherent to the CaVEMan algorithm, we used the following post-processing filtering criteria for substitutions: a minimum of two reads in each direction reporting the mutant allele, at least tenfold coverage at the mutant allele locus, minimum variant allele fraction 5%; no insertion or deletion called within a read length (150 bp) of the putative substitution, no soft-clipped reads reporting the mutant allele, and a median BWA alignment score of the reads reporting the mutant allele ≥140. The following variants were flagged for additional inspection for potential artifacts, germline contamination or index-jumping event: any mutant allele reported within 150 bp of another variant, any mutant allele with a population allele frequency >1 in 1000 according to any of five large polymorphism databases (ExAC, 1000 Genomes Project, ESP6500, CG46, Kaviar), variant reported in more than 10% of the tumor samples and mutant allele reported in >1% of the matched normal reads. For indels, the inbuilt filters of the Pindel algorithm, as implemented in our pipeline, were used. In addition, recurrent indels occurring in >2 samples were flagged for additional inspection.

Mutational signatures were derived using principal component analysis and non-negative matrix factorization as implemented in the SomaticSignatures R package[39].

**Variant validation.** The Cancer Genome Project (Wellcome Trust Sanger Institute) variant calling pipeline has been continually validated and bench-marked[40,41]. We confirmed variant calling quality through manual visual inspection of raw sequencing read for 8% of all variants called. All rearrangements reported were validated by reconstruction at base pair resolution and by cDNA reads spanning the breakpoint junction.

**Analysis of mutations in cancer genes.** We considered variants as potential drivers if they presented in established cancer genes[42]. Tumor suppressor coding variants were considered if they were annotated as functionally deleterious by an in-house version of VAGrENT (http://cancerit.github.io/VAGrENT/)[43] or were disruptive rearrangement breakpoints or focal (<1 Mb) homozygous deletions. Mutations in oncogenes were considered driver events if they were located at previously reported canonical hot spots (point mutations) or amplified the intact gene. Amplifications also had to be focal (<1 Mb) and increase the copy number of oncogenes to a minimum of five copies for a diploid genome. To search for driver variants in novel cancer genes or in non-coding regions, we employed previously developed statistical methods that identify significant enrichment of mutations, taking into account various confounders such as overall mutation burden and local variation in the mutability of the genomic region[44].

**Targeted mutation screening.** RNA from frozen tumors (1 μg) or corresponding to approximately 5 cm² of 10 μm FFPE sections was reverse transcribed using oligo-dT or random hexamer primers (RevertAid first strand cDNA synthesis kit, ThermoFisher). PCR screening was performed using primer combinations that allow amplification of candidate alterations as well as additional control fragments from the unaffected allele to assess cDNA quality. Amplified fragments were sequenced by Sanger sequencing (GATC, Konstanz, Germany) using primers detailed in Supplementary Table 1.

**Immunohistochemistry.** Immunohistochemical staining for phospho-ERK1/2 (Cell Signaling Technology, clone D13.14.4E) was performed according to standard protocol (dilution 1:800, pre-treatment with target retrieval TR6.1, Dako). Results were scored in a semi-quantitative fashion (negative, weak, moderate, strong).

**Code availability.** The algorithms used to analyze sequencing data are available at http://cancerit.github.io/.

**Data availability.** All data supporting the findings of this study are available within the article and its supplementary files or from the corresponding author on reasonable request. Sequencing data have been deposited at the European Genome-Phenome Archive (http://www.ebi.ac.uk/ega/) that is hosted by the European Bioinformatics Institute (accession numbers EGAS00001002534 and EGAS00001002171).

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

## Acknowledgements

This work was supported by funding by the Wellcome Trust, St. Baldrick's Foundation, the Deutsche Forschungsgemeinschaft (GE 539/13-1), the Deutsche Krebshilfe (50-2709-Gr2, T9/96/TrI, 50-2721-Tr2) and NIHR GOSH BRC. G.C., S.B., C.G., P.J.C., and M.R.S. received personal fellowships from the Wellcome Trust. The Cooperative Weichteilsarkom Studiengruppe (CWS) was additionally supported by the Deutsche Kinderkrebsstiftung (SoTiSaR, A2007/13DKS2009.08) and by the Förderkreis Krebskranke Kinder e.V. Stuttgart, Germany. The SIOP-RTSG/GPOH-nephroblastoma study group is supported by the charity "Elterninitiative krebskranker Kinder im Saarland e.V.". We thank children and their families for participating in our research and the clinical teams involved in their care. We thank Sabine Roth and Sharna Lunn for expert technical assistance.

## Author contributions

J.W., G.C., M.D.C.V.H., and C.G. analyzed sequencing data. C.V. performed histological analyses. S.Ba., H.S., and B.Z. provided technical assistance. S.J.F., M.J., J.A., O.S., C.D., R.F., N.G., D.T.W.J., C.K., S.M.P., W.M., E.K., N.S., A.R. and M.S.-S. curated and reviewed the samples, clinical data, and/or provided clinical expertise. M.R.S. and P.J.C. contributed to discussions. M.G. and S.B. directed this research and wrote the manuscript, with contributions from G.C., J.W., and M.D.C.V.H.

## Additional information

**Competing interests:** The authors declare no competing interests.

