## [Peer Review File · Nature Communications]

Reviewers' comments:

Reviewer #1 (Remarks to the Author):

This study examines cryptogenic soft tissue tumors of infants using WGS and RNA-Seq. There are several interesting new findings, most notably a novel ITD in EGFR and a complex duplication deletion mutation of BRAF. Both of these appear to be activating, and are important for both improved tumor classification and potentially for precision therapy of these rare tumors, particularly for CMN. The data look quite solid. There are a few points for the authors to address.

1) Table 1 presents the mutations found. There is an interesting subset of cases in the validation samples with no mutations identified. To understand exactly what this means, one has to look at Supp. Table 1 to find that only EGFR-ITD, BRAF-ID the ETV6-NTRK3 fusion were tested in those samples. That information should be included in the Table legend. In its current form, Table 1 incorrectly implies that all the various translocations included were absent in the "wildtype".

2) Abstract, first sentence. The phrase "tumors without canonical gene fusion" is a bit confusing as it does not refer to the specific diagnoses that follow, but rather samples within those diagnostic groups that lack the fusions already known to occur in those entities. Please clarify language.

3) The co-occurrence of BRAF and NTRK3 mutations in some tumors is interesting. Do mutations co-occur in same cell or could this represent two clones? Would the allele frequencies suggest one or the other of these interpretations?

4) There is no call-out for supp. table 4 but there is a call-out for a non-existent table 5.

Reviewer #2 (Remarks to the Author):

The authors describe molecular findings in two genetically and morphologically related tumors of infancy; congenital mesoblastic nephroma and infantile fibrosarcoma. 10/10 classical CMN lacking a NTRK3 fusion revealed a single intragenic in-frame internal tandem duplication of the EGFR kinase domain. This confers sensitivity to a targeted EGFR inhibitor, afatinib. Additionally 2/4 mixed cellularity CMN also harbored an EGFR ITD. Moreover mutations in the BRAF oncogene were found in 2/3 cellular histology CMNs were found. Every cryptogenic CMN interrogated contained an oncogenic rearrangement in BRAF, EGFR or NTRK1. Subsequently results were validated in a cohort of CMN, Infantile Fibrosarcoma, Wilms tumor, clear cell sarcoma, malignant rhabdoid tumors and nephroblastomatosis.

Although CMN's and IFS are relatively benign tumors, these molecular findings add to our understanding of the molecular mechanisms and may suggest potential targeted therapies in case of irresectable or recurrent lesions.

Reviewer #3 (Remarks to the Author):

Overall, this is a good study that systematically discovered biologically and clinically relevant intragenic rearrangements in EGFR and BRAF in patients lacking known driver mutations.

Major comment

While the BRAF ITD show downstream activation, this is not surprising. However, the rare, yet interesting fusions, involving NTRK gene family members should have additional validation to show

the downstream activation. Similarly, if they do represent potential therapeutic targets, some supporting data would make this more convincing.

Minor comment

Figure 1 legend is not accurate. For instance, while it reads "(c) Schematic of wildtype transcript." the actual panel C clearly shows the mutant cDNA.

Recurrent intragenic rearrangements of EGFR and BRAF in soft tissue tumors of infants

Response to reviewer's comments

We thank our Reviewers for taking the time to consider our manuscript. We are grateful for the comments which have helped us to improve this work. Below we provide a point-by-point response to the reviewers' comments.

Reviewer #1

#	Reviewer's comment	Our response
1.1	This study examines cryptogenic soft tissue tumors of infants using WGS and RNA-Seq. There are several interesting new findings, most notably a novel ITD in EGFR and a complex duplication deletion mutation of BRAF. Both of these appear to be activating, and are important for both improved tumor classification and potentially for precision therapy of these rare tumors, particularly for CMN. The data look quite solid. There are a few points for the authors to address.	We thank the reviewer for their appreciation of our study.
1.2	1) Table 1 presents the mutations found. There is an interesting subset of cases in the validation samples with no mutations identified. To understand exactly what this means, one has to look at Supp. Table 1 to find that only EGFR-ITD , BRAF-ID the ETV6-NTRK3 fusion were tested in those samples. That information should be included in the Table legend. In its current	We apologies for this lack of clarity and thank the reviewer for highlighting it. We have reconfigured Table 1 and Supplementary Table 1 to address this comment.

	form, Table 1 incorrectly implies that all the various translocations included were absent in the “wildtype”.	
1.3	2) Abstract, first sentence. The phrase “tumors without canonical gene fusion” is a bit confusing as it does not refer to the specific diagnoses that follow, but rather samples within those diagnostic groups that lack the fusions already known to occur in those entities. Please clarify language.	Thank you for highlighting this. We have expanded and clarified the introductory sentences of the abstract, which now read: ‘Soft tissue tumors of infancy encompass an overlapping spectrum of diseases that pose unique diagnostic and clinical challenges. We studied genomes and transcriptomes of cryptogenic congenital mesoblastic nephroma (CMN), and extended our findings to five anatomically or histologically related soft tissue tumors: infantile fibrosarcoma (IFS), nephroblastomatosis, Wilms tumor, malignant rhabdoid tumor and clear cell sarcoma of the kidney.’
1.4	3) The co-occurrence of BRAF and NTRK3 mutations in some tumors is interesting. Do mutations co-occur in same cell or could this represent two clones? Would the allele frequencies suggest one or the other of these interpretations?	Unfortunately, we cannot confirm whether these mutations are in the same clone or two separate clones. Only a single FFPE block was available for each of these validation cohort samples. Furthermore, the DNA was analysed by nested PCR amplification, which required different primer pairs to amplify the wildtype and mutant alleles. Hence there is regrettably no reliable means of estimating allele frequencies. Given the absence of any other apparent founding driver event, it would be particularly remarkable for these two mutations to occur independently of each other and give rise to separate clonal populations The statement in our discussion ‘the finding of co-mutation of NTRK3 and BRAF in IFS raises the possibility of intrinsic resistance of some tumor to TRK inhibition applies to both potential scenarios. Specifically, an independent BRAF-driven clone lacking an NTRK3 fusion would equally be likely to escape TRK inhibition. Hence, we advocate extending the diagnostic work up of refractory or

		relapsed CMN and IFS to whole genome sequencing, particularly in the context of clinical trials. We have changed the manuscript to highlight this possibility of two independent clones within tumors. Last paragraph of Results section: 'We were unable to accurately estimate relative allele frequencies by nested PCR (Methods). Hence, it is possible that both fusions co-exist within the same clone or represent independent clones that evolved in parallel within the same tumor bulk.' Second paragraph of Discussion: 'The finding of co-mutation of NTRK3 and BRAF in IFS raises the possibility of intrinsic resistance of some tumors to TRK inhibition, regardless of whether these mutations occur in the same clone or in independent competing clones.'
1.5	4) There is no call-out for supp. table 4 but there is a call-out for a non-existent table 5.	We thank the reviewer for highlighting this error, which has now been corrected.

Reviewer #2:

#	Reviewer's comment	Our response
2.1	The authors describe molecular findings in two genetically and morphologically related tumors of infancy; congenital mesoblastic nephroma and infantile fibrosarcoma. 10/10 classical CMN lacking a NTRK3 fusion	We thank the reviewer for their comments.

	revealed a single intragenic in-frame internal tandem duplication of the EGFR kinase domain. This confers sensitivity to a targeted EGFR inhibitor, afatinib. Additionally, 2/4 mixed cellularity CMN also harbored an EGFR ITD. Moreover, mutations in the BRAF oncogene were found in 2/3 cellular histology CMNs were found. Every cryptogenic CMN interrogated contained an oncogenic rearrangement in BRAF, EGFR or NTRK1. Subsequently results were validated in a cohort of CMN, Infantile Fibrosarcoma, Wilms tumor, clear cell sarcoma, malignant rhabdoid tumors and nephroblastomatosis. Although CMN's and IFS are relatively benign tumors, these molecular findings add to our understanding of the molecular mechanisms and may suggest potential targeted therapies in case of irresectable or recurrent lesions.	
--	--	--

Reviewer #3:

#	Reviewer's comment	Our response
3.1	Overall, this is a good study that systematically discovered biologically and clinically relevant intragenic rearrangements in EGFR and BRAF in patients lacking	We thank the reviewer for their appreciation of our work and for raising this interesting point. Indeed, in the discovery cohort we identified 2 CMN samples (evaluated by

	known driver mutations. Major comment While the BRAF ITD show downstream activation, this is not surprising. However, the rare, yet interesting fusions, involving NTRK gene family members should have additional validation to show the downstream activation. Similarly, if they do represent potential therapeutic targets, some supporting data would make this more convincing.	whole genome sequencing) with rare fusions involving NTRK1 (LMNA-NTRK1 and NTRK1-TPR) As highlighted in the original manuscript, these fusions have previously been observed in rare cases of IFS^{1,2,3} and have also been functionally assessed^{4,5,6,7}. As for evaluating whether these rare fusions represent potential therapeutic targets, following submission of our manuscript a phase I/II clinical trial of a selective NTRK inhibitor, larotrectinib, has been published in the New England Journal of Medicine⁸. Here, a durable clinical response to TRK inhibition was observed in children and adults who had NTRK1/2/3-mutant tumors. This included tumors with LMNA-NTRK1 and TPR-NTRK1 fusions. We have updated our introduction and discussion, citing this important study by Drilon et al.
3.2	Minor comment Figure 1 legend is not accurate. For instance, while it reads “(c) Schematic of wildtype transcript.” the actual panel C clearly shows the mutant cDNA.	We thank the reviewer for pointing out this error, which has been rectified in revised Figure 1.

References

1. Church AJ, *et al.* Recurrent EML4-NTRK3 fusions in infantile fibrosarcoma and congenital mesoblastic nephroma suggest a revised testing strategy. *Modern pathology: an official journal of the United States and Canadian Academy of Pathology, Inc.*, (2017).

2. Davis JL, Lockwood CM, Albert CM, Tsuchiya K, Hawkins DS, Rudzinski ER. Infantile NTRK-associated Mesenchymal Tumors. *Pediatric and developmental pathology: the official journal of the Society for Pediatric Pathology and the Paediatric Pathology Society*, 1093526617712639 (2017).
3. Tannenbaum-Dvir S, *et al.* Characterization of a novel fusion gene EML4-NTRK3 in a case of recurrent congenital fibrosarcoma. *Cold Spring Harbor molecular case studies* **1**, a000471 (2015).
4. Sartore-Bianchi A, *et al.* Sensitivity to Entrectinib Associated With a Novel LMNA-NTRK1 Gene Fusion in Metastatic Colorectal Cancer. *Journal of the National Cancer Institute* **108**, (2016).
5. Okimoto RA, Bivona TG. Tracking Down Response and Resistance to TRK Inhibitors. *Cancer discovery* **6**, 14-16 (2016).
6. Doebele RC, *et al.* An Oncogenic NTRK Fusion in a Patient with Soft-Tissue Sarcoma with Response to the Tropomyosin-Related Kinase Inhibitor LOXO-101. *Cancer discovery* **5**, 1049-1057 (2015).
7. Russo M, *et al.* Acquired Resistance to the TRK Inhibitor Entrectinib in Colorectal Cancer. *Cancer discovery* **6**, 36-44 (2016).
8. Drilon A, *et al.* Efficacy of Larotrectinib in TRK Fusion-Positive Cancers in Adults and Children. *The New England journal of medicine* **378**, 731-739 (2018).

REVIEWERS' COMMENTS:

Reviewer #1 (Remarks to the Author):

The authors have responded adequately to the previous critique.

Reviewer #3 (Remarks to the Author):

The authors had addressed all comments.